# Metal-Assisted Deprotonation as a Key Step in Selective Copper Extraction: A Theoretical and Experimental Study

**DOI:** 10.3390/ijms262411955

**Published:** 2025-12-11

**Authors:** Rene Maurelia, Pedro Pablo Zamora, Felipe M. Galleguillos Madrid, Víctor M. Jiménez-Arévalo

**Affiliations:** 1Departamento de Química y Biología, Facultad de Ciencias Naturales, Universidad de Atacama, Av. Copayapu 485, Copiapó 1530000, Chile; rene.maurelia@uda.cl; 2Centro de Desarrollo Energético Antofagasta, Universidad de Antofagasta, Antofagasta 1240000, Chile; felipe.galleguillos@uantof.cl; 3Departamento de Química de Los Materiales, Facultad de Química y Biología, Universidad de Santiago de Chile, Av. L. B. O’Higgins 3363, Casilla 40, Correo 33, Santiago 9170022, Chile; victor.jimenez@usach.cl

**Keywords:** DFT, Fukui-function, reaction intermediate

## Abstract

The growing demand for copper, together with the environmental limitations of conventional recovery methods, has intensified the search for extractants capable of operating directly in acidic mining solutions. In this work, a combined experimental–theoretical approach is presented to understand the coordination and extraction behaviour of Cu^2+^, Ni^2+^, Co^2+^ and Cd^2+^ ions with the ligand HDDMP (4-hexyl-dithiocarboxylate-5-hydroxy-3-methyl-1-phenylpyrazole). Experimental solvent-extraction tests show that copper forms stable coordination complexes even under highly acidic conditions (pH ≈ 0), unlike Ni^2+^, Co^2+^ and Cd^2+^, which require higher pH values for efficient extraction. DFT calculations reveal that Cu^2+^ promotes a spontaneous, low-barrier deprotonation–coordination process that is exergonic and electronically stabilised through strong Cu–S orbital interactions. This mechanism explains the exceptional selectivity of HDDMP towards copper, in which the copper ion acts simultaneously as both a coordinating centre and a deprotonating agent. These findings provide a molecular basis for designing new extractants suited to hydrometallurgical environments, offering direct industrial relevance for acidic copper-recovery circuits, minimising reagent consumption and improving selectivity in solvent-extraction processes widely used in mining operations.

## 1. Introduction

The growing global demand for strategic metals, particularly copper, has stimulated the development of more efficient technologies for their direct recovery from complex mineral sources such as tailings, industrial effluents, and low-grade leaching solutions. Direct metal-recovery processes, which selectively extract metal ions from an aqueous phase without requiring precipitation or electrodeposition steps, offer a sustainable and highly efficient strategy, especially in operations where selectivity and product purity are critical to process profitability.

Solvent extraction remains one of the most versatile and widely applied techniques in metallurgical processing, largely because the selectivity of the system can be finely tuned through rational extractant design, pH control, and the choice of organic phase [1,2,3]. In the copper industry, for example, extractant selectivity plays a pivotal role in the performance of direct-recovery systems, enabling discrimination between Cu^2+^ and other cations present in solution, such as Ni^2+^, Co^2+^, and Cd^2+^ that compete for the same coordination sites [4,5]. The rational design of extractant ligands with selective affinity for transition metals makes it possible to optimise the recovery of valuable metals from complex industrial solutions while reducing environmental impact and associated operational costs [6,7]. Among the extractants most employed for copper recovery from acidic media are oxime-based systems and organophosphorus compounds of the Cyanex family. Cyanex 272, 302, and 301, for instance, function as acidic extractants whose mechanism involves deprotonation of the phosphinic group followed by coordination through oxygen or sulphur, although their extraction efficiency depends strongly on pH control [8].

Despite their effectiveness, numerous studies have shown that several organophosphorus extractants exhibit high toxicity and low biodegradability, increasing their environmental footprint and posing risks to human health during handling and disposal [9,10]. In this context, HDDMP-type ligands (4-hexyl-dithiocarboxylate-5-hydroxy-3-methyl-1-phenylpyrazole), which contain mixed O/S donor atoms, offer a significant advantage: deprotonation generates a coordinating O^−^ centre, while the sulphur atom provides additional affinity for Cu^2+^, promoting the formation of stable ML_2_ complexes [11,12] even under acidic conditions. This behaviour is consistent with reports on sulphur-based extractants such as Cyanex 301, which display a greater affinity for ‘soft’ cations than their oxygen-based analogues [8]. The coexistence of O and S donor atoms within a single ligand confers selectivity, complex stability and operational capability at low pH [13], supporting the interest in HDDMP-type structures as potential alternatives to conventional extractants. Several studies have shown that the Cu^2+^ ion can induce ligand deprotonation, promoting the formation of stable complexes at the organic/aqueous interface [3,14,15]. This phenomenon, known as metal-assisted deprotonation, gives copper a distinctive coordination behaviour compared with other metals, explaining its high extractive affinity under acidic conditions [16]. Understanding this process is not only important from a theoretical standpoint but also essential for optimising direct copper-recovery processes, in which extraction efficiency depends on the balance between ligand deprotonation and effective coordination at the interface [1].

This work proposes a detailed mechanistic model, supported by DFT calculations and experimental data [13], describing how Cu^2+^ promotes HDDMP deprotonation and coordination under acidic conditions. Using computational analysis tools such as the reduced density gradient (RDG), Fukui functions [17,18], OPDOS, geometry optimisation and dual reactivity descriptors, a clear correlation is established between the local reactivity of the ligand and the electronic affinity of the metal centres involved. The results not only provide fundamental insight into the extraction mechanism but also offer a robust theoretical basis for the design of new extractants suitable for direct metal-recovery systems, particularly within the copper industry. Finally, HDDMP was modelled using the structural data reported by Maurelia et al. in [13], ensuring consistency with experimental observations.

## 2. Results and Discussion

### 2.1. Molecular and Acid–Base Characteristics of the Ligand

Figure 1a shows the full structure of the ligand along with its coordinating region. For the purposes of this study, only the coordinating fragment was considered, since the alkyl chains neither participate nor interfere with the formation of the coordination complex [19,20]. These chains were replaced with methyl groups to simplify the model and allow a more focused analysis of the coordination site and the metal–ligand complex formation mechanism.

Although the alkyl chains are bulky, they do not cause notable steric hindrance or change the complex geometry. As shown in Figure 1b, they point in opposite directions, allowing the metal-binding site to remain essentially planar.

The interaction between HDDMP and divalent metal ions (Cu^2+^, Ni^2+^, Co^2+^ and Cd^2+^) is strongly pH-dependent (Figure 2), as coordination occurs at the interface between the organic phase (Escaid 103) and the aqueous phase. Because the ligand is completely insoluble in water, the deprotonation required for metal–ligand complex formation takes place exclusively at the organic/aqueous interface during coordination [1]. In this process, the proton from the hydroxyl group (–OH) is displaced and transferred into the aqueous phase, while the resulting anion is stabilised through interaction with the metal cation. To rationalise this behaviour, the standard free energy of deprotonation of HDDMP in the organic phase was calculated, with emphasis on relative trends that reflect its propensity for deprotonation rather than absolute values. The calculation was carried out using the classical thermodynamic Relation (1):(1)$${∆G}^{0}=G_{A}+G_{H^{+}}-G_{HA};\;pKa=\frac{{∆G}^{0}}{2.303RT}$$
where $G_{A}$ is the free energy of the anionic species, $G_{HA}$ corresponds to the protonated form, and $G_{H}$ is the standard free energy of the proton in the medium. Since the calculations were carried out in a non-polar solvent, cyclohexane was chosen because its dielectric constant is similar to that of Escaid 103 (ε ≈ 2.0), and an estimated value of $G_{H}=160.13$ kcal/mol was used [4,21].

The calculated theoretical pKa (~117) should be interpreted with caution. It is important to emphasise that such an extremely large theoretical pKa for HDDMP in a non-polar phase does not represent an intrinsic measure of acidity but rather reflects well-known thermodynamic and methodological limitations associated with modelling deprotonation in low-dielectric media. In these environments, charge separation is inherently disfavoured, and continuum solvation models reproduce this by imposing a strong desolvation penalty on the conjugate base. Furthermore, proton solvation has no physical meaning in apolar solvents, and the reference term used in the thermodynamic cycle introduces a systematic upward shift in the absolute pKa scale. The absence of hydrogen-bond stabilisation, the restricted conformational reorganisation of the ligand, the lack of explicit ion-pairing with metal ions, and the known tendency of continuum models to overestimate the energy of isolated anions in low-dielectric media all contribute to the unusually high value. Consequently, the absolute pKa should not be interpreted literally; instead, it reflects the strong thermodynamic disincentive for HDDMP to deprotonate in an apolar medium. The mechanistic analysis presented in this work, therefore, relies exclusively on relative trends across the metal series, which remain fully consistent with the experimentally observed selectivity.

On the other hand, this value does not represent an acid–base equilibrium constant; rather, it reflects the very low propensity of HDDMP to deprotonate in the organic phase. This result is consistent with the fact that deprotonation does not occur spontaneously in the organic medium and must instead be promoted either by the presence of a metal ion with sufficient affinity to stabilise the anionic species at the moment of coordination, or by a sufficiently high pH at the organic/aqueous interface [1]. This explains why metal-complex formation at the interface can be pH-dependent, depending on the coordinating metal. These behaviours are fully consistent with the experimental results (Figure 2) [13], in which Cu^2+^ is extracted at pH ≈ 0, whereas Ni^2+^, Co^2+^ and Cd^2+^ require more basic conditions to form their corresponding complexes.

From the experimental extraction curves (%E vs. pH) [13], a theoretical curve (black line in Figure 2) was fitted using a standard sigmoidal model (non-linear regression) of the Henderson–Hasselbalch type [21,22,23], applied directly to the experimental extraction data (%E vs. pH) [13]. In this way, an apparent average pKa of 7.8 (Figure 2b) was estimated for the systems containing Ni^2+^, Co^2+^, and Cd^2+^, corresponding to the inflection point at which 50% extraction efficiency is achieved. This point represents the effective interfacial pH at which the ligand becomes thermodynamically available to form the metal complex, rather than its intrinsic acidity in a homogeneous solution [24,25,26]. In contrast, the Cu^2+^ ion exhibits notably high extraction efficiency even at pH values close to 0, indicating that copper possesses an exceptional ability to induce ligand deprotonation under strongly acidic conditions. The fitted model yields an apparent pKa of 1.7 (Figure 2a), evidencing a far more favourable activation of the ligand compared with the other metals. This distinctive behaviour implies that, unlike the other cations, Cu^2+^ does not require elevated pH to form the coordination complex, enabling efficient extraction even in acidic media. This observation suggests not only a higher electronic affinity but also a synergistic effect between the metal’s charge-accepting capacity and its ability to promote ligand deprotonation, indicating that copper facilitates ligand activation in acidic environments at a substantially lower energetic cost.

On the other hand, when chloroform is used as the organic phase [13], copper shows similar behaviour. At pH 3.9, Cu^2+^ extraction increases progressively with HDDMP concentration, reaching a maximum of approximately 95% at ligand concentrations above 0.025 M (Figure 3). This trend confirms that, even at moderately acidic pH, copper is able to form the coordination complex efficiently in a low-polarity organic medium. Thus, the high extraction efficiency of HDDMP towards Cu^2+^ is evident in both Escaid 103 and chloroform.

### 2.2. Kinetic, Quantum and Thermodynamic Study of HDDMP Metal Complexes

Regarding the calculated standard free energies of formation (ΔG°) for the Cu^2+^, Cd^2+^, Co^2+^ and Ni^2+^ complexes with the ligand in its anionic form (Equation (2)), the values show a clear trend fully consistent with the experimental extraction curves (%E vs. pH) presented in Figure 2a,b. The Cu^2+^ ion displays the highest thermodynamic affinity, with a ΔG° of −29.5 kcal·mol^−1^ when coordinated to two HDDMP ligands. This value contrasts sharply with those obtained for Cd^2+^ (−2.4 kcal·mol^−1^), Co^2+^ (−3.5 kcal·mol^−1^) and Ni^2+^ (−4.6 kcal·mol^−1^), confirming that Cu^2+^ is extracted efficiently even at low pH [14,15]. See Table 1.(2)$${∆G}^{0}=G_{complex}^{0}-\left(G_{metal}^{0}-G_{ligant}^{0}\right)$$

When the free ligand (without metal) is in its protonated form (Figure 1), the main nucleophilic centres are located on the sulphur and oxygen atoms. This is evidenced by the dual reactivity descriptor, which shows blue regions, indicative of nucleophilic character around these atoms, with the sulphur atom contributing more strongly than the hydroxyl oxygen (Figure 4a). The dual descriptor is a DFT-based tool that compares the electron density after the addition or removal of one electron, enabling the identification of regions that behave as nucleophilic or electrophilic sites.

In addition, the RDG analysis (Figure 4b) reveals a stabilising interaction between the sulphur atom and the hydroxyl hydrogen, indicating that this hydrogen is spatially constrained through an intramolecular S···H interaction prior to metal coordination. The RDG (Reduced Density Gradient) method is a graphical approach used to detect non-covalent interactions by analysing spatial variations in the electron density. In these plots, the green isosurfaces correspond to regions where such interactions occur, allowing them to be readily visualised and interpreted.

From this initial configuration, formation of the Cu–ligand complex under acidic conditions proceeds through two sequential transition states (TS1 and TS2), separated by a metastable intermediate. This intermediate presents an energy barrier of approximately 4 kcal/mol toward TS1 and 2 kcal/mol toward TS2 Figure 5. See Table 1.

The IRC analysis describes how the energy of the system evolves as the reaction progresses from reactants to products via the transition state. In simple terms, the IRC profile indicates the energy required at each stage of the process and provides insight into the reaction rate. A lower energy barrier at the maximum of the pathway corresponds to a more accessible transition state and therefore a faster reaction, whereas a higher barrier indicates a less favourable transition state and a slower process.

In Figure 5, the first step of the copper extraction mechanism in acidic medium corresponds to the interaction and subsequent deprotonation of the hydroxyl group (–OH) of the ligand, marking the onset of metal–ligand complex formation. This proton displacement leads to the characteristic interactions associated with the first transition state (TS1), illustrated in Figure 6a. Simultaneously, the hydrogen undergoes a rotation of approximately 180°, adopting a position further from Cu^2+^ and beginning to move away from the oxygen atom, as evidenced by the RDG isosurface shown in Figure 6a. As a result, the O–H interaction gradually loses its covalent character (Figure 6a and Figure 7).

As shown in Figure 1 and Figure 4, when the ligand is in its metal-free form, the proton of the hydroxyl group preferentially orients towards the sulphur atom, establishing a stabilising intramolecular interaction. When the proton is forced to rotate through 360°, the resulting energy profile (Figure 7) shows that disrupting this stabilised orientation and inducing rotation requires approximately 20 kcal·mol^−1^, depending on the metal interacting with the ligand. However, in the presence of Cu^2+^, strong repulsion between the metal centre and the proton allows the system to reach 180°, leading to deprotonation of the –OH group and interrupting the continuity of the rotational curve (Figure 7). This behaviour indicates that, in the case of copper, formation of the intermediate complex is not hindered whether the ligand is in its protonated or deprotonated form. In contrast, the other metals evaluated do not exert a significant influence on the orientation of the hydroxyl group, which remains essentially unchanged relative to the metal-free ligand [27]

The OPDOS analysis provides insight into the evolution of the system’s electronic structure by illustrating how the orbitals of the ligand and the metal overlap along the energy axis. The OPDOS curve indicates whether this overlap results in bonding, non-bonding or anti-bonding interactions. Positive values reflect stabilising orbital combinations, whereas negative values correspond to antibonding contributions that oppose complex formation. This information helps assess how favourable or unfavourable a given coordination step may be: a stronger and more extended bonding region generally correlates with a more stable interaction, while predominant antibonding character denotes a weaker and less favourable coordination environment. Consequently, OPDOS analysis clarifies not only the nature of the interaction but also the extent to which the ligand stabilises the metal centre across different energy levels.

The OPDOS profile in Figure 6b shows that both the Cu–S and Cu–O interactions are only partially stabilised, reflecting the intermediate nature of the transition state. In the energy region below the Fermi level (EF), weak bonding contributions are observed, whereas near EF a significant antibonding density dominates, indicating that these interactions have not yet developed into fully stabilised bonds.

The Cu–S interaction (yellow curve) displays a moderate antibonding band close to EF, suggesting that overlap between the copper d orbitals and the sulphur p orbitals is already partially established, giving rise to an attractive interaction that is still in the process of forming. By contrast, the Cu–O interaction (blue curve) shows a much weaker bonding contribution and a more pronounced antibonding region near EF, indicating a less efficient overlap and a more incipient coupling.

The RDG analysis (Figure 6a) supports these observations, revealing green isosurfaces between the Cu centre and the sulphur atom, characteristic of an emerging attractive interaction. In contrast, the RDG regions associated with Cu–O are weaker, as the hydrogen still retains some degree of interaction with the oxygen while moving away. These findings indicate that delocalisation of electronic density towards the metal centre has begun, although it is not yet sufficient to yield a fully stabilised coordination environment. Furthermore, the dual reactivity descriptor shows that the –OH group possesses lower nucleophilicity than sulphur, which contributes to the difference in interaction strength between these two atoms (Figure 4 and Figure 6).

Partial release of the proton alters the local electronic environment, increasing the donor character of the oxygen atom; in other words, the nucleophilic region associated with the –OH group becomes more pronounced (Figure 8c), facilitating charge redistribution between O and S. As a result, the system exhibits cooperative interactions in which sulphur contributes more strongly to the emerging coordination bond, while oxygen gradually increases its participation as deprotonation proceeds [28,29] (Figure 6a).

Once copper has been stabilised through coordination with the ligand (Figure 8a) and the proton has been fully removed from the oxygen atom, the interactions between Cu and the ligand strengthen, as shown by the RDG plots in Figure 8a. Consequently, TS1 decreases in energy and evolves into a more stable bidentate intermediate with Cu^2+^. The OPDOS analysis further indicates that the Cu–S curve exhibits a substantially larger bonding region than Cu–O (Figure 8b), demonstrating that, in the ground state, the Cu–S interaction involves stronger orbital coupling. The intermediate also displays a smaller antibonding peak compared with TS1, located just above the Fermi level. This energetic proximity suggests that, upon electron gain or under modified redox conditions, the antibonding orbitals of the Cu–S system could be readily populated, consistent with a metastable species. In contrast, the Cu–O system presents a lower density of bonding states and a reduced antibonding density near the Fermi level compared with TS1, resulting in a weaker interaction than Cu–S in the ground state. In the RDG plots, this is manifested as more localised and less extensive attractive interactions (Figure 8b).

The presence of these empty states in antibonding regions indicates a strong tendency of the system to accept electrons, a characteristic feature of reactive species that have not yet reached an electronically optimal, fully stabilised state [30]. Under such conditions, the formation of new bonds is favoured, particularly with nucleophilic species present in the medium. The dual descriptor further confirms that the copper atom defines the preferred sites for electrophilic attack, highlighted in green in Figure 8c. As noted previously, once the proton is removed, the nucleophilic reactivity at the oxygen site increases, thereby facilitating its interaction with copper. The energy barrier between the intermediate and TS2 is approximately 2 kcal·mol^−1^, meaning that the intermediate readily evolves into TS2. Consequently, it will quickly react with another free ligand, whether in its protonated or deprotonated form, at the organic/aqueous interface. As shown in Figure 9a,b,d, the copper atom in the intermediate approaches the sulphur atom and, despite carrying only a single positive charge, is still capable of generating a repulsive effect on the hydrogen of the OH group, analogous to the earlier step, ultimately promoting its departure. As observed in Figure 9a, the dual descriptor displays behaviour similar to that of TS1, as the nucleophilicity of the protonated oxygen is considerably lower than that of sulphur. Thus, as the proton is progressively displaced, the oxygen regains its nucleophilic reactivity, enabling the coordination interaction to proceed [31].

As shown in Figure 9a,b,d, the copper atom in the intermediate approaches the sulphur atom and, despite carrying only a single positive charge, is still capable of exerting a repulsive effect on the hydrogen of the OH group. This behavior mirrors the previous step and ultimately promotes proton departure. As illustrated in Figure 9a, the dual descriptor displays behaviour analogous to that observed for TS1, as the nucleophilicity of the protonated oxygen remains considerably lower than that of sulphur. Consequently, as the proton is progressively displaced, the oxygen regains its nucleophilic reactivity, enabling the coordination interaction to proceed [31]. In this context, the dual descriptor highlights the copper centre as an electrophilic region (green surface) not simply because of its formal charge, but because the descriptor evaluates the change in electron density upon electron removal. This quantity reflects how strongly a site attracts additional electron density, and even a Cu^+^ centre can appear markedly electrophilic when its local electronic environment is polarised by the surrounding ligand framework. Thus, the green region around copper indicates that it acts as an electron-accepting site along the reaction pathway, consistent with its role in forming the Cu–S and Cu–O coordination bonds.

The interactions present in TS2 exhibit a much stronger character (Figure 9b) compared with those in TS1 (Figure 6a). This increase in interaction strength is also influenced by the generation of additional bonding contributions as the system approaches a more advanced stage of coordination.

The presence of antibonding states below the Fermi level does not necessarily imply immediate bond destabilisation; rather, their effect depends on the balance between bonding and antibonding contributions, as well as the degree of orbital overlap. A bond can remain stable if the bonding interactions outweigh the antibonding ones. This balance—clearly reflected in the OPDOS profiles—enables a coherent interpretation of the electronic structure and its correlation with the observed stability [32].

The OPDOS curves for the Cu–S and Cu–O interactions (Figure 9c) reveal pronounced differences in the nature and extent of orbital interactions. In the Cu–S contribution, the presence of a well-defined positive lobe below the Fermi level corresponds to bonding interactions between the copper and sulphur orbitals. These bonding states indicate the initial stabilisation of the Cu–S interaction during complex formation. Just above the Fermi level, the same curve shows a small negative contribution associated with the onset of antibonding character in the emerging tetradentate complex. The relatively low magnitude of this feature suggests that, at this stage, the destabilising effect of the antibonding interactions is insufficient to outweigh the strong bonding contributions below the Fermi level [33]. In contrast, the OPDOS profile corresponding to Cu–O is almost flat (Figure 9c), displaying only a very weak positive lobe at low energies and an almost negligible antibonding signal above the Fermi level. This confirms that oxygen contributes very little to the electronic overlap at this stage of the mechanism.

About the frontier orbitals of the coordination intermediate (Figure 9d), these are predominantly of d-character and begin to overlap with the frontier orbitals of the incoming sulphur atom. This emerging orbital overlap explains the appearance of bonding contributions just below the Fermi level in the Cu–S curve, which stabilise the interaction. In contrast, the Cu–O interaction still does not exhibit significant orbital overlap, consistent with the minimal bonding contributions observed in the OPDOS profile (Figure 9c) [34].

In Figure 10, the RDG plots of the final copper complex are shown. The isosurfaces appear in the central region of the complex, indicating strong non-covalent interactions localised between the copper atom and the coordinating groups. Although these interactions do not reach the strength of covalent bonds, they are sufficiently intense to contribute to the stability of the complex and to its copper-extracting capability, preventing structural collapse. This property may be advantageous in mining processes.

Once the complex is stabilised—that is, in its final form—the OPDOS analysis (Figure 11) shows that the Cu–S interaction exhibits more intense bonding contributions below the Fermi level, accompanied by a corresponding antibonding region above it. This behaviour arises from the close energetic proximity between the copper d orbitals and the sulphur 3p orbitals, which enhances orbital overlap and consequently strengthens both the bonding stabilisation and the appearance of its antibonding counterparts [34]. By contrast, the Cu–O interaction displays slightly weaker bonding contributions than Cu–S, along with a reduced antibonding signal, owing to the more limited orbital overlap. This indicates that oxygen participates in the coordination in a secondary manner, providing electrostatic stabilisation and additional interactions, but with a less covalent character than sulphur. In contrast, the remaining metals (Cd^2+^, Co^2+^ and Ni^2+^) show a much stronger dependence on the degree of ligand deprotonation. Their ΔG° values, which are considerably less negative, indicate that complex formation is far more sensitive to the availability of the anionic form of the ligand. This behaviour is reflected experimentally in the need to reach higher pH values to achieve comparable extraction efficiencies, suggesting that in these cases the acid–base equilibrium at the interface constitutes the rate-limiting step for complex formation [15,16].

Although Co^2+^ and Cd^2+^ may adopt higher coordination numbers in aqueous media, our DFT optimisations show that, in the presence of two HDDMP molecules, the M–HDDMP complexes consistently converge to approximately planar tetracoordinated geometries in both the aqueous PCM model and the organic-phase model. This reflects the strong O,O-chelating nature of HDDMP, which effectively saturates the most favourable coordination sites of the metal and prevents the stabilisation of additional axial ligands under extraction conditions. The behaviour observed for Cu and Co is consistent with their stronger electron-correlation character within the HDDMP ligand environment.

The comparison between experimental and theoretical data supports a model in which extraction efficiency depends not only on the intrinsic acidity of the ligand, but also on the metal’s ability to induce ligand deprotonation and coordination at the interface. This phenomenon is particularly pronounced in the case of Cu^2+^, whose highly favourable ΔG° allows the equilibrium to shift even under conditions where the anionic form of the ligand is scarcely available. In contrast, for Cd^2+^, Co^2+^ and Ni^2+^, complex formation becomes thermodynamically viable only once the pH has sufficiently promoted prior ligand deprotonation [14,15].

This indicates that Cu^2+^ acts not only as a coordinating species but also as a deprotonating agent, which explains its extraction efficiency even under conditions where the fraction of anionic ligand is minimal. This distinctive property accounts for why the behaviour of Cu^2+^ cannot be satisfactorily fitted to the nonlinear regression used for the other metals, clearly highlighting its anomalous character in selective extraction processes [15].

When comparing the IRC profiles for each metal (Figure 12a), it is evident that the formation of the HDDMP_2_–M complexes (M = Ni, Co, Cd) proceeds through a single-step mechanism, as shown by the presence of only one transition state in each case. This suggests that the two donor centres of the ligand—oxygen and sulphur—must coordinate with the metal almost simultaneously, without any detectable stable intermediates.

The OPDOS results (Figure 12b) show that, in the Ni complexes, the Ni–O and Ni–S curves exhibit broad bonding regions below the Fermi level and shallow antibonding minima. This pattern reflects efficient orbital overlap between the metal d orbitals and the ligand donor orbitals, giving rise to a strongly stabilised covalent interaction. The minimal antibonding contribution indicates an optimal distribution of electronic density around the metal centre, consistent with the high structural stability of the complex.

In the case of Co, the bonding contributions remain significant, but more pronounced antibonding lobes appear in the region close to the Fermi level. As a result, the overall stability of the Co complex is intermediate between that of Ni and Cd [34,35].

Cd, by contrast, shows the weakest bonding profiles and the largest antibonding contributions above the Fermi level, indicating less efficient orbital coupling. Because Cd^2+^ has a fully occupied d^10^ configuration, its interaction with the ligand donor atoms is predominantly electrostatic, with minimal covalent character. This leads to more labile coordination and lower stabilising electron density around the metal centre [34,35].

The RDG analysis reveals further differences in the pattern and extent of non-covalent interactions among the metal complexes. The Ni complex shows the most extensive isosurfaces in the coordination region, indicating a dense network of attractive interactions. This behaviour can be attributed to the compact nature of the Ni^2+^ 3d orbitals, which promotes efficient orbital overlap with the ligand donor atoms, favouring multiple electrostatic and dispersion-type contacts. These interactions generate a region of cooperative electronic delocalisation which, although not equivalent to full covalent bonding, contributes substantially to the stability of the complex [36]. In contrast, the Co complex displays moderate and more localised isosurfaces, indicative of fewer and more spatially concentrated non-covalent interactions, as well as a slightly more stabilised electronic environment in the final state.

Conversely, the Cd complex exhibits the least extensive and lowest intensity isosurfaces, reflecting limited orbital overlap. The lower effective electron density of Cd^2+^ and the weak participation of its fully occupied 4d orbitals reduce its ability to establish attractive interactions with the ligand, resulting in a more diffuse and energetically less favourable coordination environment [35].

“The optimized structures indicate that although the Cd and Co complexes shown in Figure 13b,c may appear planar depending on the viewing angle, their actual geometries are clearly distinct. Cadmium adopts a distorted tetrahedral arrangement, whereas cobalt exhibits a planar coordination environment. This flattening in Co arises from its d^7^ electronic configuration, for which axial ligand stabilization is considerably weakened in a low-dielectric medium, while in-plane donor interactions and π-type contributions become comparatively more favorable, energetically driving the system toward a compressed, planar structure.”

In this context, the cations shown in Figure 13 do not exhibit any deprotonating ability, whereas copper does. This distinction is particularly significant because Cu^2+^ can promote deprotonation even under acidic conditions, where it remains fully soluble and readily available to form stable complexes. This represents an important industrial advantage, as copper is typically processed in acidic media, such as during leaching operations, meaning that the ligand can capture it without the need to modify the chemical environment [37,38,39]. As a result, a selective window emerges that is fully compatible with the operational conditions commonly encountered in copper-processing systems. Thus, the deprotonation step not only explains how the complex is formed but also clarifies why this mechanism can be realistically applied in hydrometallurgical settings.

## 3. Materials and Methods

All calculations were performed using the Gaussian 09 software package. Geometry optimisations were carried out with the ω-B97X-D [40] density functional in combination with the cc-pVTZ basis set for all atoms—a level of theory that offers balanced accuracy for both structural and electronic properties in molecular systems containing transition-metal centres. Harmonic vibrational frequency calculations at the same level were used to characterise the stationary points and to obtain the Gibbs free energies. Transition states (TSs) were identified as first-order saddle points with a single imaginary frequency, and intrinsic reaction coordinate (IRC) calculations were performed in both directions to trace the reaction pathway and confirm the connection with the corresponding minima. Further electronic-structure analyses—including reduced density gradient (RDG), overlap population density of states (OPDOS), and the dual reactivity descriptor—were carried out using the Multiwfn 3.8 program [22] based on the Gaussian output files.

## 4. Conclusions

This study demonstrates that the selectivity of the HDDMP ligand toward Cu^2+^ arises from a metal-assisted deprotonation mechanism in which copper acts simultaneously as a coordinating centre and a deprotonating agent. DFT calculations reveal a two-step coordination process involving two transition states with very low activation barriers (Ea ≈ 2–4 kcal·mol^−1^), together with a highly favourable free energy of formation (ΔG° = −29.5 kcal·mol^−1^). This behaviour stands in sharp contrast to the much less negative ΔG° values calculated for Ni^2+^, Co^2+^, and Cd^2+^.

Combined RDG and OPDOS analyses show that the Cu–S interaction dominates the electronic stabilisation of the complex, while oxygen contributes secondary electrostatic interactions. Accordingly, the proposed model links thermodynamic affinity to the capacity of the metal to promote ligand activation, establishing a mechanistic basis for deprotonation and complexation that can guide the design of selective extractants for practical application in copper metallurgical processes.

## Figures and Tables

**Figure 1 ijms-26-11955-f001:**
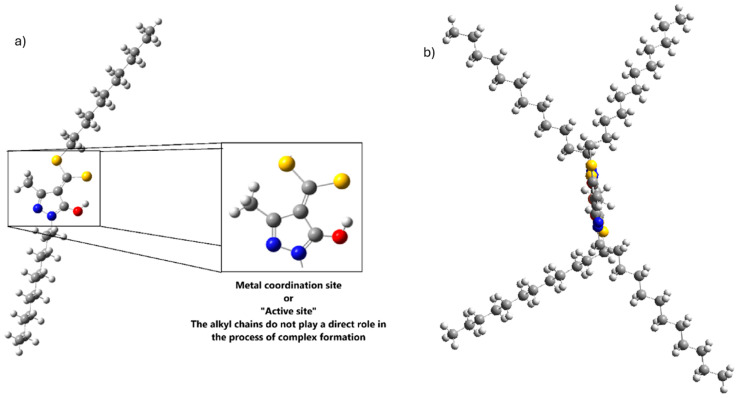
(**a**) Structure of the HDDMP ligand and its coordinating site. (**b**) Lateral view of the complex. S (ball yellow), O (ball red), N (ball blue), C (ball gray), and H (ball white).

**Figure 2 ijms-26-11955-f002:**
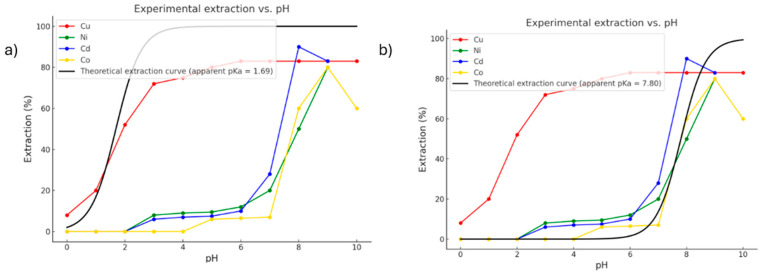
Experimental metal extraction data [13]. (**a**) Sigmoidal fit for Cu^2+^ R^2^ = 0.92 (**b**) Sigmoidal fits for Co^2+^, Ni^2+^, and Cd^2+^ R^2^ = 0.95. Red line = Cu, green line = Ni, Blue line = Cd, yellow line = Co, black line = theoretical extraction.

**Figure 3 ijms-26-11955-f003:**
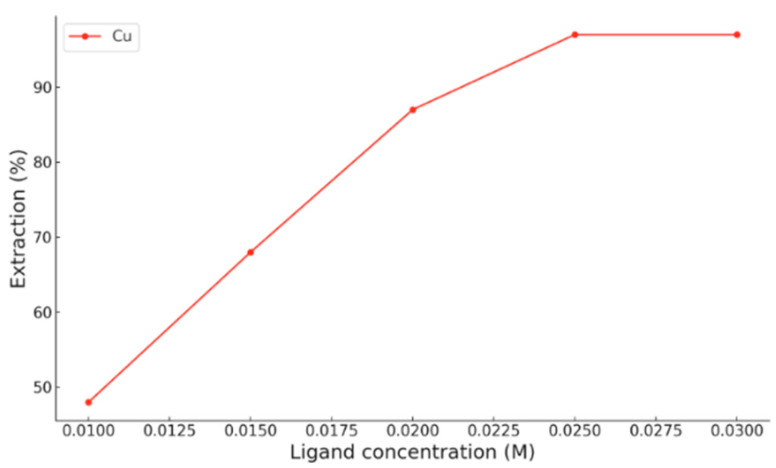
Experimental extraction of Cu^2+^ at different HDDMP concentrations in chloroform at pH 3.9 [13].

**Figure 4 ijms-26-11955-f004:**
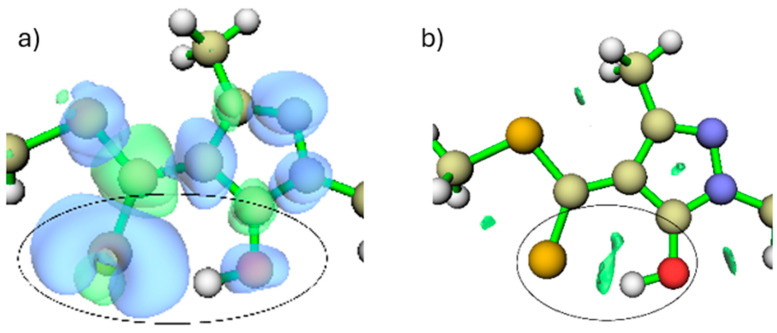
(**a**) Dual reactivity descriptor isosurface (blue = nucleophilic, green = electrophilic; isovalue = 0.05). (**b**) RDG isosurface showing the intramolecular S···HO interaction. The green surfaces inside the circle identify the region where this attractive interaction occurs, which stabilizes the position of the hydroxyl oxygen in this conformation. (isovalue = 0.05).

**Figure 5 ijms-26-11955-f005:**
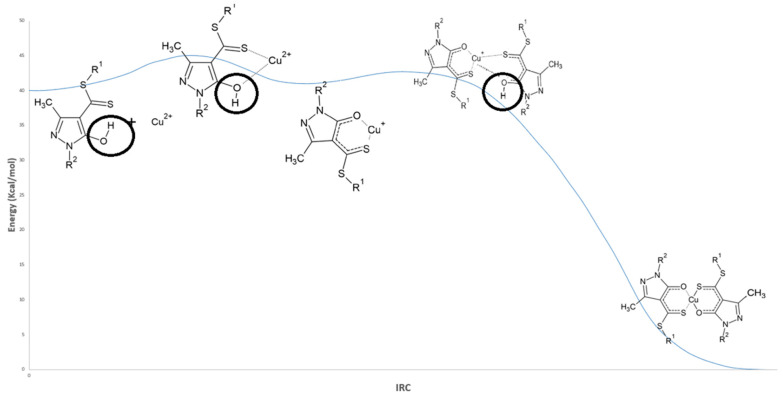
IRC profile for the formation of the Cu complex in acidic medium (HDDMP_2_–Cu). R_1_ = R_2_ = –CH_3_.

**Figure 6 ijms-26-11955-f006:**
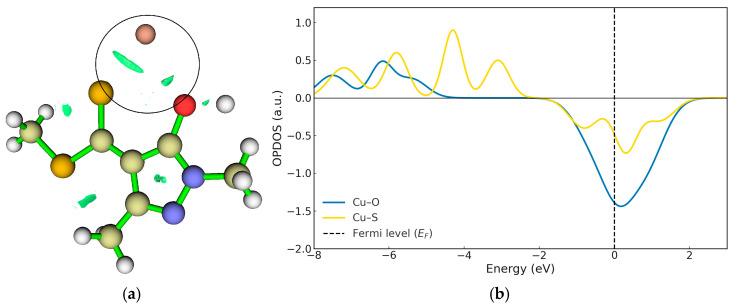
(**a**) RDG interaction plots for TS1. (**b**) OPDOS Cu–S (Yellow line) and Cu–O (blue line) for TS1. The OPDOS curve (in atomic units) represents the degree of orbital overlap between the selected fragments as a function of energy, where positive values indicate bonding interactions, negative values indicate antibonding interactions, and values near zero correspond to nonbonding character.

**Figure 7 ijms-26-11955-f007:**
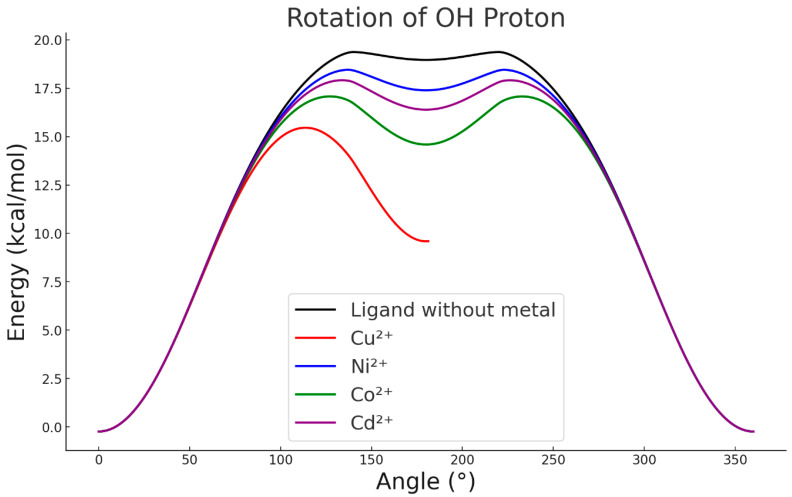
OH rotational scan with different metals.

**Figure 8 ijms-26-11955-f008:**
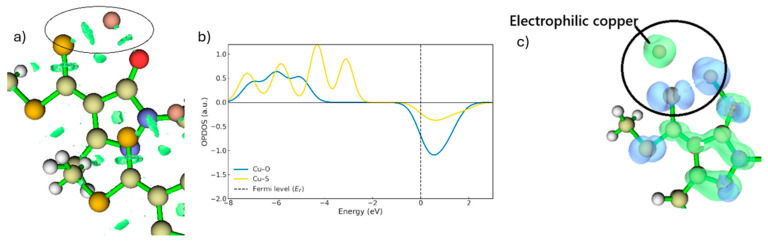
(**a**) RDG regions for the Cu–O and Cu–S interactions are shown inside the circle (isovalue = 0.05). (**b**) OPDOS for the intermediate. (**c**) Dual reactivity descriptor for the intermediate (blue represents nucleophilic regions, green represents electrophilic regions. Isovalue = 0.05).

**Figure 9 ijms-26-11955-f009:**
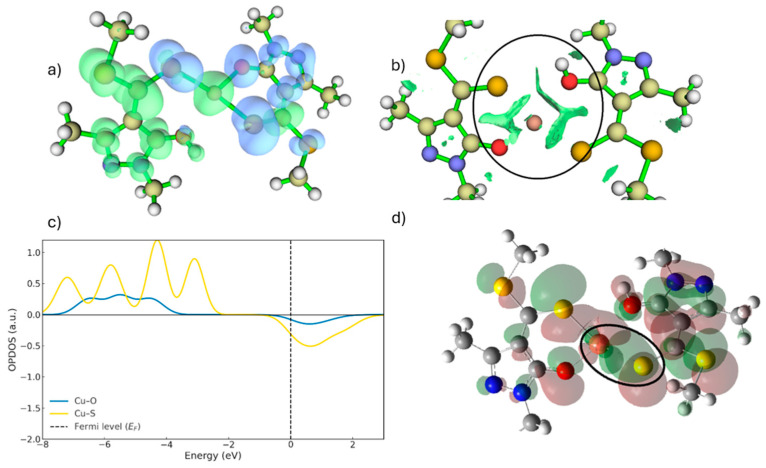
(**a**) Dual descriptor for TS2. (**b**) RDG for TS2 (Inside the circle, pronounced green RDG surfaces appear, revealing strong S–Cu and O–Cu interaction regions). (**c**) OPDOS for TS2. (**d**) Frontier orbital overlap of the ligands in TS2.

**Figure 10 ijms-26-11955-f010:**
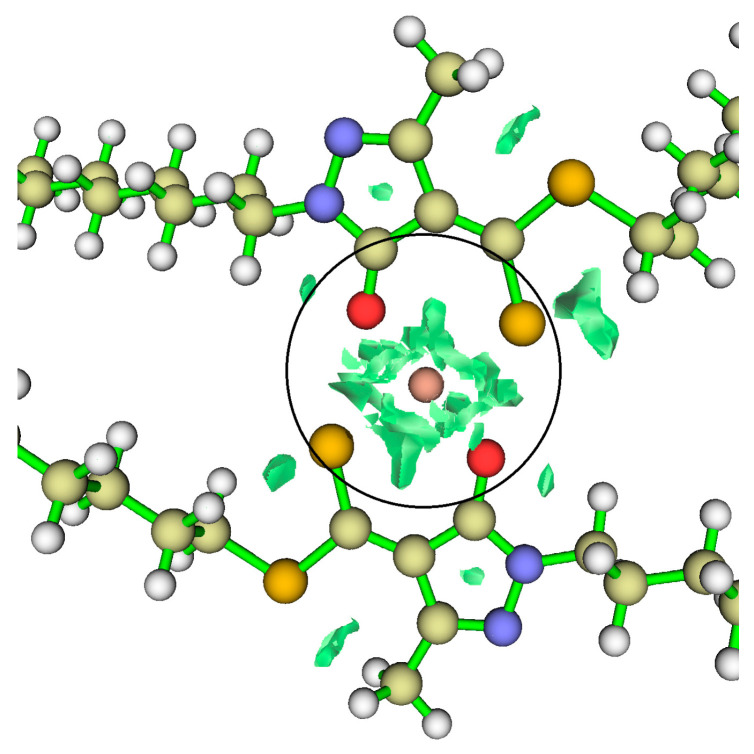
RDG plots for the final (stable) HDDMP_2_–Cu complex. Inside the circle, the copper atom is shown, and the green RDG isosurfaces surrounding it indicate intense noncovalent interactions in the final complex.

**Figure 11 ijms-26-11955-f011:**
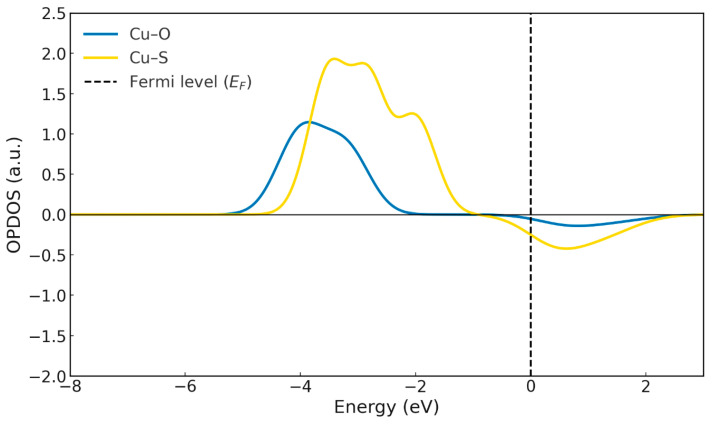
OPDOS for the Cu-HDDMP_2_ complex. Cu-O (Blue line) Cu-S (Yellow line).

**Figure 12 ijms-26-11955-f012:**
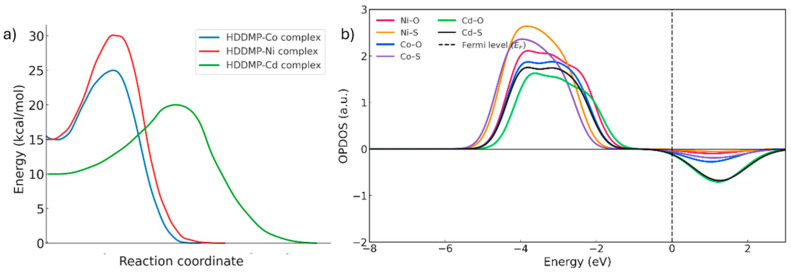
(**a**) IRC for the metals (blue line = Co, Red line = Ni, Green line = Cd). (**b**) OPDOS of the bonds in the divalent metal complexes (HDDMP_2_–M^2+^). Red line = Ni-O, orange line = Ni-S, blue line = Co-O, purple line = Co-S, green line = Cd-O, black line = Cd-S.

**Figure 13 ijms-26-11955-f013:**
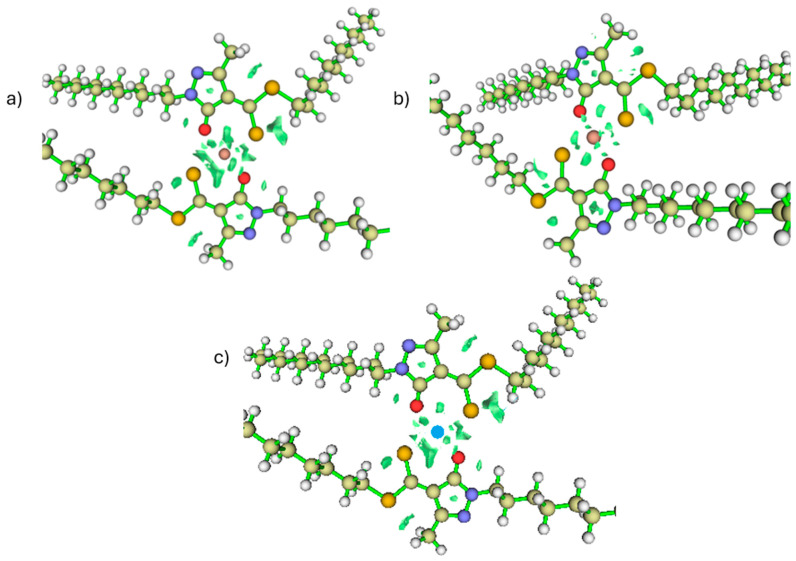
RDG (**a**) Nickel. (**b**) Cobalt. (**c**) Cadmium.

**Table 1 ijms-26-11955-t001:** Summary of ΔG° formation energies and TS energies for M-HDDMP^2^ complexes.

Metal	Complex/STATE	ΔG° (kcal/mol)	Intermediate (kcal/mol)	TS1 (kcal/mol)	TS2 (kcal/mol)
Cu^2+^	Cu-HDDMP_2_ (2 ligands)	−29.5	6	8	10
Ni^2+^	Ni-HDDMP_2_ (2 ligands)	−4.6	-	15	-
Co^2+^	Co-HDDMP_2_ (2 ligands)	−3.5	-	10	-
Cd^2+^	Cd-HDDMP_2_ (2 ligands)	−2.4	-	8	-

## Data Availability

The data presented in this study are available on request from the corresponding author.

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
