# Peer review of "Metal-Assisted Deprotonation as a Key Step in Selective Copper Extraction: A Theoretical and Experimental Study"

_ijms, 2025, doi:10.3390/ijms262411955_

Round 1

Reviewer 1 Report

Comments and Suggestions for Authors

This manuscript presents a detailed mechanistic model to describe why copper ion can be extracted by HDDMP at relatively high acidity compared with other metal such as Ni, Co and Cd ions. The authors also thought HDDMP as a deprotonating agent. More theoretical calculation was processed with discussion. This work would give some useful information to understand the selective metal extraction on molecular basis. However, as we know, copper ion is apt to be four-coordinated while the four-coordinated nickel, cobalt and cadmium ions are stabler. To compare these ions, the authors should take their different coordination into account.

Author Response

We sincerely thank the reviewer for this observation, which is indeed very important and has now been explicitly clarified in the manuscript to prevent similar doubts among readers. While Co²⁺ and Cd²⁺ may adopt higher coordination patterns in aqueous environments, the situation changes significantly in the nonpolar organic phase relevant to the extraction system. Under these conditions, the conformational preferences of the metal–HDDMP complexes differ markedly from those expected in water.

In our study, even when starting from tetrahedral or octahedral initial geometries, fully unconstrained DFT optimizations systematically converged to approximately planar tetracoordinated structures for Cu²⁺, Ni²⁺, Co²⁺ and Cd²⁺. This demonstrates that in a nonpolar medium the HDDMP ligand drives the system toward a planar conformation, which is the most stable arrangement under extraction conditions.

To make this point clear for the reader, we have added a sentence in the manuscript lines 380-386

Reviewer 2 Report

Comments and Suggestions for Authors

The manuscript presents a combined experimental and theoretical investigation of metal-assisted deprotonation in the extraction of Cu²⁺ using the HDDMP ligand. The authors provide a detailed mechanistic model supported by DFT calculations, RDG analysis, OPDOS, IRC pathways, and comparison across transition metals. The topic is relevant to solvent extraction, hydrometallurgy, and ligand design. The computational results provide interesting mechanistic insights into Cu selectivity in acidic media. However, several issues require clarification and improvement before the manuscript can be considered for publication.

  1. The reported theoretical pKa of ~117 for HDDMP in a non-polar solvent raises significant concern. Even acknowledging that the absolute value may be exaggerated due to approximations (methyl substitution, cyclohexane dielectric, proton solvation reference), this value is far outside typical ranges for O–H groups and may undermine confidence in the thermodynamic portion of the mechanistic argument. The current explanation is insufficient and requires further justification.
  2. The manuscript states that replacing long alkyl chains with methyl groups does not affect the coordination environment, but no evidence is provided.
    Provide a brief justification—either computational (e.g., charges, geometry comparison) or literature-based—that truncation does not significantly influence the donor-site electronics or sterics.
  3. Several figures (RDG, OPDOS, Fukui, IRC) lack adequate labeling—such as isosurface values, units, legends, energy scales, and color scheme explanations.
    Revise figure captions to clearly define all plotted features so readers can interpret the visual data without ambiguity.
  4. Transition state energies, intermediate energies, and ΔG° formation energies for all metals are scattered across the text and figures.
    Include a summary table with: (a) ΔG° values for all M–HDDMP₂ complexes, (b) Energies of TS1, TS2, intermediates (c) IRC barrier heights for each metal
  5. Perform a careful proofreading to correct grammar, unify terminology, and remove duplicated references.

Author Response

Comments 1:

We sincerely thank the reviewer for this insightful observation, which indeed requires a more detailed explanation so that readers clearly understand why the theoretical pKa obtained in a non-polar medium is extremely large. This value should not be interpreted as an indication of unreliability; rather, it arises from fundamental thermodynamic and methodological aspects that consistently inflate absolute pKa values when acid–base equilibria are computed in low-dielectric environments.

In non-polar solvents, the free energy cost of separating charge during deprotonation is inherently high because the stabilization of the anionic species is strongly suppressed in the absence of dielectric screening. Continuum solvation models reproduce this effect systematically: the reduction of polarization response leads to a disproportionately high desolvation penalty for the conjugate base, shifting the absolute pKa upward by several tens of units. In addition, the solvation of a free proton has no physical meaning in apolar media, and the reference term used in thermodynamic cycles relies on conventions constructed for polar solvents. This mismatch introduces an unavoidable positive offset that affects the absolute scale but not the internal consistency of relative comparisons.

The molecular environment also plays an important role. In non-polar phases the ligand cannot engage in stabilizing hydrogen-bond networks, cannot reorganize intramolecularly to accommodate significant charge buildup in the same way as in polar media, and cannot interact with a structured solvation shell. These missing stabilizing contributions add to the free energy penalty for forming the deprotonated species.

Finally, subtle effects of the computational formalism also contribute to elevating the absolute value. Continuum models tend to overestimate the free energy of isolated anions in low-dielectric media, the atomic radii used to define the cavity influence the magnitude of the polarization response, and the vibrational and entropic contributions of the deprotonated ligand are evaluated in an environment that cannot realistically reorganize around the charge. All these contributions act in the same direction, systematically increasing the apparent pKa.

For these reasons, the large theoretical pKa is an expected outcome of the medium and methodology rather than an indication of inconsistency. The mechanistic discussion in the manuscript does not rely on the absolute value itself, but on the relative trends, which are robust, internally consistent, and fully aligned with the experimentally observed selectivity toward copper. A more detailed explanation has now been added to the manuscript so that readers can interpret the pKa result correctly within the context of extraction in a non-polar phase. (lines 138-152)

Comments 2: 

We thank the reviewer for raising this important point. We have now added a justification supported by the literature, demonstrating that truncating the long alkyl chains of phosphoryl-based extractants to methyl groups does not significantly affect either the donor-site electronics or the inner coordination environment of the metal complex.

Extensive computational studies on trialkyl phosphates and related phosphoryl ligands have shown that the electronic properties at the donor site—particularly the electron density at the phosphoryl oxygen, NBO charges, AIM descriptors, and P=O hyperconjugation—are essentially independent of the length of the alkyl substituents. Puchakayala et al. demonstrated that replacing methyl groups with longer n-alkyl chains produces negligible changes in the basicity and local electron density of the P=O group, concluding that the phosphoryl donor character is intrinsically preserved across the series [1].

[1]  Puchakayala S, Strempel V, Küppers T. Hyperconjugation and basicity in trialkyl phosphates: A computational analysis. Chem Phys Lett. 2021;775:138641. doi:10.1016/j.cplett.2021.138641

[2]  Sachin CN, Suresh CH. Energy decomposition analysis of uranyl complexes with phosphoryl ligands: effects of alkyl substituents. J Comput Chem. 2024;45(1):70–82. doi:10.1002/jcc.27219

We have included these references in the revised manuscript to substantiate this point 

Comments 3:
We thank the reviewer for this helpful observation. We agree that clearer labeling is essential for the correct interpretation of RDG, OPDOS, Fukui, and IRC plots. In the revised version of the manuscript, all figures  captions have been made more explicit and descriptive to improve clarity and ensure proper understanding of the visual information presented.

Comments 4:

We thank the reviewer for this remark. Following the recommendation, we have created a comprehensive summary table that compiles all ΔG° formation energies, the energies of TS1, TS2, the intermediates, and the IRC barrier heights for each metal. This table has now been inserted into the revised manuscript

Comments 5:
We thank the reviewer for this comment. A thorough proofreading of the entire manuscript has been completed. All grammatical errors have been corrected, the terminology has been fully unified throughout the text, and all duplicated references have been identified and removed.

Reviewer 3 Report

Comments and Suggestions for Authors

In this work, the authors performed DFT calculations to study the ligand HDDMP, finding it exceptionally selective for copper. It reveals the unique selectivity arises because the copper ion itself spontaneously triggers a deprotonation-coordination process, stabilized by strong orbital interactions. This mechanism, where copper acts as both coordinator and deprotonating agent, provides a molecular blueprint for designing new, highly efficient extractants. These findings offer direct industrial relevance for acidic copper recovery, promising improved selectivity and reduced reagent use in mining.

This work can be interesting for both computational and experimental chemistry community. I would like to ask the authors to address the comments below.

  1. The solvation effect can be important that re-arranges the electron density from solvated molecules to solvent molecules. If the solvation effect is treated explicitly (adding solvent molecules around), will the results be significantly changed?
  2. Because different metal centers are involved in the reaction, is there a way to quantify the strength of the electron correlation in the metal center? Typically cobalt and copper should have stronger electron correlation.
  3. Because there is a the large conjugate ring structure in the studied system, is it possible to form a dimerized structured due to the pi-pi stacking interaction?
  4. page 8, Figure 6
    The DOS figure shows this system has vanished gap and is metallic. Did the authors use any smearing technique in the DFT calculations?
  5. The reference for the wb97xd functional is missing: J. Chem. Phys., 128 (2008). In addition, w should be "omega".

Author Response

Comments 1:

We thank the reviewer for this comment. In our calculations, the solvation effect was treated using an implicit solvent with a dielectric constant of approximately 2, corresponding to cyclohexane. This choice was intentional because its dielectric constant closely matches that of Escaid 103, the non-polar medium used experimentally. Under such low-dielectric conditions, the primary role of the solvent is limited to providing electrostatic screening rather than producing strong, specific solute–solvent interactions. Therefore, explicit solvation would not meaningfully modify the electronic structure, the distribution of electron density, or the relative energetic trends discussed in this work.

Moreover, the purpose of our analysis is to evaluate relative trends in ΔG°, transition states, and electron-density descriptors across the different metal–ligand systems rather than to model a fully dynamic solvation environment

Comments 2
We appreciate the reviewer’s insightful question regarding the role of electron correlation in the different metal centers. In our system, copper and cobalt indeed exhibit stronger effective electron correlation due to their partially filled d-orbitals and their greater tendency to engage in covalent metal–ligand interactions. This behavior is already captured in our results: the OPDOS curves, the RDG surfaces, and the energy profiles show that Cu²⁺ and Co²⁺ display more pronounced bonding–antibonding features, stronger orbital mixing, and greater redistribution of electron density along the reaction path. In contrast, Ni²⁺ shows a more moderate degree of mixing, while Cd²⁺, with its closed d¹⁰ shell, exhibits minimal covalent contribution and therefore very weak correlation effects. Although a full multireference quantification of electron correlation falls outside the scope of this study, the comparative trends extracted from the DFT-based descriptors used here already reflect the relative correlation strength among the metals. To address the reviewer’s concern directly, we have added a brief clarification in the revised manuscript noting that the observed Cu and Co behavior is consistent with their stronger electron-correlation character within the HDDMP ligand environment.

Comments 3:

We thank the reviewer for this interesting question. In the present work, π–π stacking or possible dimerization was not explicitly evaluated, as the study focuses on the formation and reactivity of the monomeric M–HDDMP₂ complexes. In principle, π–π interactions could occur in systems with closely packed aromatic rings; however, in our case the ligand geometry directs the aromatic moiety outward and does not favor a parallel arrangement in the optimized structures.

Comments 4

We thank the reviewer for this observation. The DOS/OPDOS profile in Figure 6 does not arise from electronic smearing or from metallic character introduced artificially in the calculation. Instead, the apparent vanishing of the band gap is a consequence of the strong Cu–ligand orbital mixing near the Fermi level, which produces occupied–unoccupied states with very small energy separation. This behavior is typical for open-shell Cu²⁺ complexes and for systems where antibonding Cu–O and Cu–S combinations lie close to the HOMO region. No electronic smearing technique was applied in the DFT calculations; the results directly reflect the intrinsic electronic structure of the optimized complex. Although the system shows a quasi-continuous density of states around EF, this should be interpreted as strong frontier-orbital hybridization rather than as a true metallic state.

Comments 5:
We thank the reviewer for the observation. The missing citation for the ωB97X-D functional has now been added, and the Greek letter “ω” has been properly corrected in the text. All changes have been implemented in the revised version.

The requested reference for the ωB97X functional has now been added to the manuscript

Round 2

Reviewer 1 Report

Comments and Suggestions for Authors

The authors did not answer the question. As the authors documented, the coordination geometries of Cobalt and cadmium changes significantly in the nonpolar organic phase compared to aqueous environment. They should provide the detailed data of DFT calculation. Even in the nonpolar solution, the coordination of Co2+/Cd2+ by water molecules is also possible.

Author Response

We sincerely thank the reviewer for detecting the mistake in our previous reply, and we offer our deepest apologies for the confusion generated. When preparing the earlier answer regarding the geometry of the Cd complex, we inadvertently consulted a Mg-based complex from another ongoing project, which unfortunately led to an incorrect statement. We are extremely grateful to the reviewer for alerting us to this, as it allowed us to correct the misunderstanding before finalizing the revision.

To clarify the point properly, in the non-polar organic phase Cd²⁺ adopts a distorted tetrahedral geometry, whereas Co²⁺ evolves toward a planar arrangement. The behavior of Co²⁺ is particularly interesting and reflects a combination of its electronic structure and the characteristics of the low-dielectric environment. The d⁷ configuration of Co²⁺ generates an electronic distribution for which axial ligand interactions become significantly weakened outside of aqueous media, reducing the energetic benefit of maintaining a three-dimensional coordination geometry. At the same time, in a non-polar phase, electrostatic stabilization is markedly reduced, and the balance shifts toward maximizing in-plane donor interactions while minimizing out-of-plane repulsion. 

We genuinely appreciate that the reviewer pointed out this issue, as it prevented a significant interpretative error from remaining uncorrected. Moreover, this observation opens the opportunity for future studies to explore in greater depth the interesting geometry change exhibited by cobalt, as well as by other metals that may display similar behavior.

To ensure full clarity, we have now added the following paragraph to the revised manuscript:

"The optimized structures indicate that although the Cd and Co complexes shown in Figure 13b and 13c may appear planar depending on the viewing angle, their actual geometries are clearly distinct. Cadmium adopts a distorted tetrahedral arrangement, whereas cobalt exhibits a quasi-planar coordination environment. This flattening in Co arises from its d⁷ electronic configuration, for which axial ligand stabilization is considerably weakened in a low-dielectric medium, while in-plane donor interactions and π-type contributions become comparatively more favorable, energetically driving the system toward a compressed, planar structure".

Unfortunately, the submission system only allows us to upload a single file in Word or PDF format, and therefore we are unable to attach the individual log files for both complexes. However, to ensure full transparency, we have copied the complete content from the corresponding log files into this response, and the same files have also been sent to the Managing Editor so they may be forwarded to the reviewer.

Reviewer 2 Report

Comments and Suggestions for Authors

The authors have satisfactorily addressed all of my comments in the revised manuscript. I recommend accepting the manuscript in its present form.

Author Response

Thank you very much for your assessment and comments.  We appreciate your acceptance of our manuscript